# Coagulation Mechanism and Compressive Strength Characteristics Analysis of High-Strength Alkali-Activated Slag Grouting Material

**DOI:** 10.3390/polym14193980

**Published:** 2022-09-23

**Authors:** Mingjing Li, Guodong Huang, Yi Cui, Bo Wang, Binbin Chang, Qiaoqiao Yin, Shuwei Zhang, Qi Wang, Jiacheng Feng, Ming Ge

**Affiliations:** 1School of Civil Engineering and Construction, Anhui University of Science and Technology, Huainan 232001, China; 2Engineering Research Center for Geological Environment and Underground Space of Jiangxi Province, Jiangxi Institute of Survey and Design, Nanchang 330015, China; 3Institute of Environment-Friendly Materials and Occupational Health, Anhui University of Science and Technology, Wuhu 241003, China; 4Hefei Comprehensive National Science Center, Institute of Energy, Hefei 230031, China; 5Wuhu Urban Construction Group Co., Ltd., Wuhu 241000, China; 6Hefei Binhu Investment Holding Group Co., Ltd., Hefei 230091, China

**Keywords:** grouting material, alkali activation, slag, mineral crystal structure, ultra-deep mine, condensation mechanism

## Abstract

In deep coal mining, grouting reinforcement and water blockage are the most effective means for reinforcing the rock mass of extremely broken coal. However, traditional cement grouting materials are not suitable for use in complex strata because of their insufficient early mechanical strength and slow setting time. This study innovatively proposes using alkali-activated grouting material to compensate for the shortcomings of traditional grouting materials and strengthen the reinforcement of extremely unstable broken coal and rock mass. The alkali-activated grouting material was prepared using slag as raw material combined with sodium hydroxide and liquid sodium silicate activation. The compressive strength of specimens cured for 1 d, 3 d, and 28 d was regularly measured and the condensation behavior was analyzed. Using X-ray diffraction and scanning electron microscopy, formation behavior of mineral crystals and microstructure characteristics were further analyzed. The results showed that alkali-activated slag grouting material features prompt and high strength and offers the advantages of rapid setting and adjustable setting time. With an increase in sodium hydroxide content, the compressive strength first increased (maximum increase was 21.1%) and then decreased, while the setting time continued to shorten. With an increase in liquid sodium silicate level, the compressive strength increased significantly (and remained unchanged, maximum increase was 35.9%), while the setting time decreased significantly (and remained unchanged). X-ray diffraction analysis identified the formation of aluminosilicate minerals as the main reason for the excellent mechanical properties and accelerated coagulation rate.

## 1. Introduction

With expanding mining scope and increasing mining depth, extremely unstable broken coal and rock mass are more frequently encountered [1]. In the original discontinuous medium with its various weak planes, the rock stratum is in equilibrium before excavation [2]. After the roadway is driven, the original equilibrium state is broken, and a large amount of energy is released, which produces deformation. The weak surface of the original coal and rock mass is further expanded, resulting in the formation of new holes and fissures [3]. Due to differences among rock properties, service life, and section size (especially the influence of in situ stress on coal and rock mass), the methods for controlling coal and rock mass deformation also differ [4]. At present, two main methods are used to improve the stability of coal and rock mass: one method seeks to improve the structure and its performance, and the other method seeks to introduce reasonable support measures [5]. However, for extremely fragile coal and rock mass, the cost of support measures is high and the effect is not significant [6].

Coal–rock mass grouting refers to the use of hydraulic, pneumatic, or electrochemical principles to inject slurry into the coal–rock mass where it can evenly gel into cracks or pores through the grouting pipe [7]. This slurry drives away the air and water in the cracks of the coal and rock mass by means of infiltration, filling, and splitting. When the slurry contains gel, the original loose rock mass and any cracks are cemented into a whole, forming a new stone body [8]. Since the 1980s, the application of grouting reinforcement technology for coal and rock maintenance applications has attracted much attention. Chinese and foreign scholars have conducted in-depth and extensive research on grouting materials. For example, Shu [9] studied the mix proportion of fiber polymer composite-reinforced cement-based grouting materials. The results showed that the mechanical properties (including compressive strength and flexural strength) of fiber polymer composite grouting materials were significantly improved, compared with ordinary grouting materials. Jiang [10] used high-performance grouting materials to solve the problem of floor heaves in broken soft rock. The results showed that active mineral composite additives can effectively improve the strength of soft rock and control floor heaves. Zhang [11] used ordinary Portland cement (PC) as the main material, and successfully prepared cement-based composite slurry to overcome the unsatisfactory reinforcement effect of grouting in broken rock masses. Cui et al. [12] prepared a high-performance grouting material suitable for water-rich silty fine sand formations using water glass as a base material and diacid ester as a curing agent. The results showed that this grouting material features adjustable setting time, high strength, and good durability.

Although there is extensive research on grouting materials, it mainly focuses on cement as the main grouting material [13], which is improved by composite fiber and other mineral additives [14,15]. Under challenges of high ground pressure, high water pressure, high ground temperature, and the complex geological structure of ultra-deep well aquifer rock, cement-based grouting materials often cannot meet engineering needs in terms of setting time, mechanical properties, and consolidation properties [16,17]. This prompted scholars to develop high-performance grouting materials to maintain the stability of the surrounding rock mass. Slag alkali-activated material is a new inorganic non-metallic material, which has developed rapidly in recent years [18,19]. Under the same preparation conditions, slag alkali-activated materials have better mechanical properties, faster setting rate, and higher early strength than cement-based materials [20,21]. Therefore, the application of slag alkali-activated materials for surrounding rock grouting of ultra-deep wells (with high ground pressure, high water pressure, and complex geological structures) may solve the technical problems associated with surrounding rock reinforcement of water-bearing strata having numerous crossings and strong construction disturbance effects [22,23].

This paper creatively proposes to apply slag alkali-activated materials to grouting materials. This approach can give full play to the characteristics of quick setting, early strength, and high strength of alkali-activated materials, and compensate for the shortcomings of traditional cement grouting materials in performance, in order to achieve improved reinforcement of extremely unstable broken coal and rock mass. The compressive strength development and condensation behavior of slag grouting materials are analyzed, mainly focusing on the formation and transformation characteristics of mineral crystals in grouting materials. The application of slag alkali-activated materials for the surrounding rock grouting of deep coal mines is an innovation that fully highlights the performance advantages of slag alkali-activated materials.

## 2. Experimental Materials and Methods

### 2.1. Experimental Materials

#### 2.1.1. Portland Cement (PC)

The study used 42.5 grade PC produced by Anhui Conch Cement Co., Ltd. (Wuhu, China). The specific surface area of PC was 336 m^2^/kg, and the particle fineness reaching the margin of a square hole sieve of 45 µm was less than 27%. Its performance conformed to common PC (GB175-2020) [24]. The chemical composition of PC tested by X-ray fluorescence spectrometry (XRF) is shown in Table 1. PC is high in calcium, and low in both silicon and aluminum cementitious material.

#### 2.1.2. Granulated Blast-Furnace Slag (GBFS)

S95 grade GBFS was purchased from Hebei Chengming Mineral Products processing plant (Shijiazhuang, China). The specific surface area of GBFS was 400 m^2^/kg, and the particle fineness reaching the margin of a square hole sieve of 45 µm was less than 19%. Its performance conformed to GBFS used in cement and concrete (GB/T 18046-2017) [25]. The chemical composition of GBFS is also shown in Table 1. Compared with PC, the content of calcium, silicon, and aluminum in GBFS is more balanced, which is more conducive to the formation of hydrated calcium silicate and hydrated calcium silicoaluminate gels as well as to the development of mechanical properties.

#### 2.1.3. Alkali Activator

For cement grouting materials, liquid sodium silicate (Na_2_SiO_3_; LSS) and distilled water were mainly used as solvents, both of which were purchased from Nanjing Bor Chemical Products Co., Ltd. (Nanjing, China). The LSS used in this work was of industrial grade with a purity greater than 95%. Na_2_O, SiO_2_, and H_2_O accounted for 9.68%, 25.26%, and 65.02% of the total mass, respectively. LSS had a modulus of 2.7 with 37 degrees Baumé (◦B’e). For alkali-activated grouting materials, sodium hydroxide, LSS, and distilled water were mainly used as solvents. Sodium hydroxide was of analytical purity and tap water was used for the test. The specific dosage of activator in the experiment is listed in Table 2.

### 2.2. Experimental Methods

#### 2.2.1. Experimental Mix Proportion

Specimen L-1 completely consisted of cement as the main raw material, mixed under free water conditions, and was used as the experimental control group. Specimens L-2 and L-3 used slag as a partial replacement of cement to test the improvement of compressive strength and setting behavior of grouting materials after slag addition. In specimens L-4 to L-6, the amount of LSS was gradually increased while maintaining a liquid–solid ratio of 0.6, and the influence of LSS on compressive strength and condensation behavior was analyzed. Specimen L-7 was a slag alkali-activated grouting material, which was only activated by sodium hydroxide. This specimen was used to study the compressive strength and setting behavior of slag grouting material and compare it with cement-based materials. In specimens L-8 and L-9, the content of sodium hydroxide was gradually increased to study the effect of sodium hydroxide on the compressive strength and condensation behavior of alkali-activated materials. In specimens L-10 to L-12, the content of LSS was gradually increased to study the effect of LSS on the compressive strength and condensation behavior of alkali-activated materials. The influence of mineral crystal characteristics and both the formation and transformation behavior of grouting materials on compressive strength and condensation behavior were analyzed.

#### 2.2.2. Preparation and Curing of Specimens

Laboratory preparations of cement-based and GBFS alkali-activated grouting materials were carried out according to the standard of testing method of cements—determination of strength (ISO method, GB/T 17671-2021) [26]. The preparation method of cement-based grouting material (specimen L-6) is described as follows: First, PC and GBFS were evenly mixed (NJ-160A cement paste mixer, Hebei Xingji Instrument Equipment Co., Ltd., Shijiazhuang, Hebei, China) according to the mix proportion in Table 2. Then, the test water and LSS were added, and rapid mixing was continued for 60 s. The mixture was then poured into a cylindrical mold with a diameter of 50 mm and a height of 100 mm. The preparation method of GBFS alkali-activated grouting material (specimen L-10) is described as follows: First, sodium hydroxide was dissolved in experimental water according to the mix proportion shown in Table 2. GBFS and sodium hydroxide solution were mixed, followed by rapid mixing for 30 s, then LSS was added and rapid mixing continued for 30 s. The mixture was poured into a cylindrical mold of the same specifications as above. The specimens were placed in the curing room immediately after preparation, and the mold was removed after 24 h of curing. After mold removal, the specimen remained in the curing room for further curing. The temperature was maintained at 20 ± 2 °C and the humidity exceeded 95%.

#### 2.2.3. Compressive Strength Test

A DYE-2000 press (obtained from Jinan Zhongchuang Industrial Test System Co., Ltd., Jinan, Shandong, China) was used to test the compressive strength of PC and GBFS alkali-activated grouting materials cured for 1 d, 3 d, and 28 d. The influence of different proportions on the development of compressive strength was analyzed. The arithmetic mean of six replicates was calculated.

#### 2.2.4. Condensation Behavior Test

The condensation behavior of PC and GBFS alkali-activated grouting materials was tested using the standard Vicat instrument (Xingjian Instrument Equipment Co., Ltd., Cangzhou, Hebei, China). The test was carried out in reference to the test methods for water requirements of normal consistency, setting time, and soundness of PC (GB/T 1346-2011) [27]. The influences of raw materials on the coagulation characteristics of PC and GBFS alkali-activated grouting materials were analyzed.

#### 2.2.5. X-ray Diffraction (XRD) Analysis

XRD was conducted using a Smart Lab SE intelligent X-ray diffractometer (RIGAKU, Tokyo, Japan) to analyze the mineral crystal composition in PC and GBFS alkali-activated grouting materials. The influences of the formation and transformation of mineral crystals on the development of compressive strength and condensation behavior were analyzed.

#### 2.2.6. Scanning Electron Microscopy (SEM) Analysis

The microstructure of paste specimens L-1, L-5, L-8, and L-11 (cured for 1 d) was analyzed with a scanning electron microscope (FlexSEM 1000, Hitachi, Tokyo, Japan). The bonding characteristics of microparticles, the structural characteristics of polymerization products, and the effects of different raw materials on the microstructure were observed under 3000 times magnification.

## 3. Experimental Results and Discussion

### 3.1. Compressive Strength Analysis

#### 3.1.1. Compressive Strength of PC Grouting Materials

Figure 1 shows the compressive strength development of PC and GBFS alkali-activated grouting materials at different curing ages. The compressive strength of specimen L-1 (100% PC, Figure 1a) reached 5.4 MPa (1 d) and further increased to 11.2 MPa (3 d). The early compressive strength of specimen L-1 (1 d and 3 d) was clearly insufficient, and the early strength developed slowly, severely affecting the early reinforcement effect of PC grouting materials [28]. Although the compressive strength further increased to 24.3 MPa when curing lasted for 28 d, this still could not compensate for the poor early reinforcement effect of PC grouting materials.

The compressive strength of specimen L-2 (80% PC and 20% GBFS, Figure 1a) decreased by 9.3% (1 d), 9.8% (3 d), and 12.7% (28 d) compared with specimen L-1. Moreover, the compressive strength of specimen L-3 (60% PC and 40% GBFS) decreased further, and the reduction range also increased, reaching between 20% and 30% compared with specimen L-1. This shows that the addition of GBFS to PC could not promote the hydration reaction and accelerate the formation of hydration products [29]. It merely played the role of inert filling and GBFS could not fully utilize its reactivity in the PC-based environment [30]. Therefore, the compressive strength of PC grouting material followed a clear decreasing trend with an increase in GBFS content.

The compressive strength of specimen L-4 (60% PC and 40% GBFS, 100 g LSS, Figure 1a) began to increase slowly but only increased by 5.4% (1 d), 5.7% (3 d), and 5.9% (28 d) compared with specimen L-3 (without LSS, Figure 1a). Moreover, although the compressive strengths of specimens L-5 (60% PC and 40% GBFS, 200 g LSS, Figure 1a) and L-6 (60% PC and 40% GBFS, 300 g LSS, Figure 1a) still followed an increasing trend, the range of the increase gradually decreased with further increase in LSS content. This shows that LSS can promote the hydration of PC and increase the production of hydration products, but the enhancement effect was very limited. LSS can provide silicon for the hydration reaction, which is an important part of hydrated calcium silicate (mainly produced by cement hydration) [31]. Therefore, increasing the silicon content is bound to also increase the formation of hydrated calcium silicate gel, thus promoting the growth of compressive strength.

#### 3.1.2. Compressive Strength of GBFS Alkali-Activated Grouting Materials

The compressive strength of specimen L-7 (Figure 1b, 100% GBFS, 20 g sodium hydroxide) reached 9.9 MPa (1 d) and further increased to 16.2 MPa (3 d) and 25.1 MPa (28 d), which are 83.3%, 44.6%, and 3.3% higher than specimen L-1, respectively (Figure 1a, PC, without sodium hydroxide). The early (1 d) compressive strength of GBFS alkali-activated grouting materials (specimen L-7) was significantly higher than that of PC grouting materials (specimen L-1). However, with extended curing age (3 d), the strength advantage of alkali-activated grouting materials decreased significantly. After the full 28 d of curing, the compressive strength of alkali-activated grouting materials was only slightly higher than that of PC grouting materials. Therefore, GBFS alkali-activated grouting materials possessed remarkable early strength characteristics, endowing them with excellent bonding performance at an early age and clear advantages over PC grouting materials [32].

The compressive strength of specimen L-8 (Figure 1b, 100% GBFS, 40 g sodium hydroxide) was significantly higher—by 14.7% (1 d), 27.8% (3 d), and 21.1% (28 d)—than that of specimen L-7 (20 g sodium hydroxide). The increase in sodium hydroxide content led to a continuous increase in the compressive strength of GBFS grouting materials, which exerted a more significant effect on the compressive strength after further curing (i.e., 3 d and 28 d). With an increase in sodium hydroxide content, the pH of the reaction solution increased significantly. The highly alkaline environment is more conducive to the depolymerization of calcium, silicon, and aluminum inside GBFS, and promotes the polymerization of active calcium, silicon, and aluminum [33]. Consequently, high-strength polymer products are formed, such as hydrated calcium silicate and hydrated calcium aluminosilicate, which promote the increase in compressive strength.

However, the compressive strength of specimen L-9 (Figure 1b, 100% GBFS, 60 g sodium hydroxide) showed a significant decreasing trend, decreasing by 12.1% (1 d), 11.6% (3 d), and 10.5% (28 d) compared with specimen L-8 (40 g sodium hydroxide). Although the highly alkaline environment is more conducive to the depolymerization and repolycondensation of active substances in GBFS, the higher sodium hydroxide content also significantly increases the content of Na^+^ in the reaction solution [34]. Na^+^ is more likely to react with the depolymerized active aluminum and silicon from GBFS to form a low-strength N-A-S-H gel. This N-A-S-H gel formation drastically affects the polycondensation reaction of active calcium with silicon and aluminum, resulting in decreased formation of C-S-H and C-A-S-H gels and a subsequent reduction in compressive strength [35]. Therefore, with an increase in sodium hydroxide content, the compressive strength of GBFS grouting material increased first and then decreased; 40 g was found to be the best dosage.

#### 3.1.3. Effect of LSS on Compressive Strength

The compressive strength of specimen L-10 (100 g LSS, 40 g sodium hydroxide, Figure 1b) increased significantly, and was 18.1% (1 d), 17.9% (3 d), and 19.1% (28 d) higher than that of specimen C-8 (without LSS). The addition of LSS further improved the compressive strength of GBFS grouting materials. However, this effect was completely different from the slight increase in compressive strength when LSS was added to the PC grouting material. This also shows that LSS can achieve a better coupling effect with GBFS grouting materials. In a suitable alkaline environment (40 g sodium hydroxide), the silicon in LSS polymerizes easily with the calcium and aluminum dissolved from GBFS to form C-S-H and C-A-S-H gels [36]. Their formation improves the reaction efficiency and the formation of polymerization products, thus promoting the improvement of compressive strength.

The compressive strength of specimen L-11 (200 g LSS, 40 g sodium hydroxide, Figure 1b) continued to increase. Compared with specimen L-10 (100 g LSS), compressive strength of specimen L-11 increased by 19.7% (1 d), 15.6% (3 d), and 11.9% (28 d). The formation of polymer products such as C-S-H and C-A-S-H gels requires silicon. However, the content of silicon in GBFS was only 32.42% (see Table 1). Insufficient silicon content prevents the repolymerization of calcium, silicon, and aluminum and also hinders the formation of polymer products [37]. The content of silicon dissolved in the reaction solution further increased with an increase in LSS, which solved the problem of insufficient silicon, and led to a significant increase in the compressive strength of specimen L-11.

However, the compressive strength of specimen L-12 (300 g LSS, 40 g sodium hydroxide, Figure 1b) did not increase significantly (only about 3% compared with specimen L-11), but it also did not decrease markedly. This indicates that the compressive strength of GBFS grouting material cannot be further improved by adding LSS over 200 g. Although the silicon content in the reaction environment increased significantly with an increase in LSS content, the formation of silicate and aluminosilicate minerals requires not only silicon, but also calcium and aluminum [38]. However, calcium only comes from GBFS (33.79%, Table 1). As the GBFS content does not increase further, the content of calcium in the reaction environment cannot increase. This significantly hinders the formation of silicate and aluminosilicate minerals and explains the significant increase in the compressive strength of specimen L-12.

### 3.2. Condensation Behavior Analysis

#### 3.2.1. Condensation Behavior of PC Grouting Materials

The setting behavior of PC grouting material is shown in Figure 2a. The initial setting time (IST) of specimen L-1 was 39 min, while the final setting time (FST) reached 326 min. Due to the slow setting and insufficient early strength development, PC grouting materials cannot provide good reinforcement in the early stage. Moreover, the IST and FST of specimen L-2 (Figure 2a) increased significantly with the addition of 200 g GBFS. The IST of specimen L-3 (Figure 2a) further increased to 64 min while the FST reached 394 min, when GBFS was increased to 400 g. GBFS cannot accelerate or promote the hydration reaction of PC; it can only function as an inert filling. Research showed that GBFS can only realize the depolymerization and repolycondensation in a highly alkaline environment (pH greater than 14), to enable polymerization, but the highest alkalinity that PC grouting materials can achieve is too low (the pH is only 13.1) [39]. Therefore, with an increase in GBFS content, the IST and FST of specimen L-3 continuously lengthened and the compressive strength decreased continuously.

The IST and FST of specimen L-4 (Figure 2a, 100 g LSS) shortened to 26 min and 35 min, which is 59.4% and 91.1% shorter than the IST and FST of specimen L-3, respectively, when 100 g LSS was added. In combination with the compressive strength results, this shows that LSS can significantly accelerate the setting of PC grouting materials, but it cannot significantly improve the compressive strength. Consequently, it cannot change the shortcomings of the poor early reinforcement effect of PC grouting materials. The IST and FST of specimens L-5 (200 g LSS) and L-6 (300 g LSS) continued to decrease with an increase in LSS content but the reduction rate decreased significantly and was only about 10%. While continuously increasing the LSS content can accelerate the setting of PC grouting materials, it still cannot significantly improve the compressive strength.

#### 3.2.2. Condensation Behavior of GBFS Grouting Materials

The condensation behavior of GBFS alkali-activated grouting material, which was completely different from that of PC grouting materials, is shown in Figure 2b. The IST of specimen L-7 (Figure 2b, 20 g sodium hydroxide) was 44 min, which is slightly longer than that of specimen L-1 (PC grouting material). However, the FST of specimen L-7 was only 69 min, which is 281 min faster than that of specimen L-1. This shows that the development rate of early mechanical properties of GBFS alkali-activated grouting materials was significantly faster compared with PC grouting materials. The hydration reaction of cement requires the initial hydrolysis of cement particles, the dissolution of active substances into the reaction solution, the formation of the initial C-S-H gel, the growth of hydration products, and the development of microstructures [40]. Therefore, the early strength of PC grouting materials developed slowly. The reaction process of alkali-activated grouting materials can only be completed through the depolymerization and repolycondensation of active substances in a highly alkaline environment [41]. Therefore, the application of alkali-activated materials in grouting materials can fully utilize its early strength advantage.

The IST and FST of specimen L-8 (Figure 2b, 40 g sodium hydroxide) further shortened to 32 min and 47 min, which is 27.3% and 31.9% lower than that of specimen L-7, respectively. Moreover, the IST and FST of specimen L-9 (Figure 2b, 60 g sodium hydroxide) further shortened to 25 min and 31 min, respectively. The setting time of GBFS grouting material continued to accelerate with an increase in sodium hydroxide content, which was more conducive to its early reinforcement performance. The improvement of the alkaline environment provides a higher alkali potential energy, improves the depolymerization rate of active substances from GBFS, and further accelerates the polycondensation of silicate and aluminosilicate minerals [42]. Thus, the condensation is accelerated and the mechanical properties are improved.

#### 3.2.3. Effect of LSS on Condensation Behavior

The IST and FST of specimen L-10 (Figure 2b, 100 g LSS) shortened to 17 min and 19 min, which is 32% and 38.7% lower than the IST and FST of specimen L-9 (without LSS), respectively. The shortening of setting time indicates that the reaction rate of specimen L-10 was further accelerated and the early mechanical properties were further improved. The underlying reason is that LSS can only accelerate the setting of PC grouting materials but it cannot significantly improve their mechanical properties. However, LSS can not only accelerate the setting of GBFS grouting materials, but it can also promote the further development of their mechanical properties, both emphasizing the characteristics of rapid setting, early strength, and high strength [43]. Therefore, the excitation effect of LSS in GBFS alkali-activated grouting materials is significantly better than that of PC grouting materials.

The IST and FST of specimen L-11 (Figure 2b, 200 g LSS) both further shortened to 14 min, which is 17.6% and 26.3% lower than the IST and FST of specimen L-10, respectively. The increased LSS content further accelerated the condensation of the specimen and the development of mechanical properties. Most importantly, the rapid development of mechanical properties is beneficial for microstructure development and results in a denser microstructure [44]. This was confirmed by the fact that the IST and FST of specimen C-11 were both 14 min, which shows that the development of structural strength was very fast.

However, the IST and FST of specimen L-12 (Figure 2b, 300 g LSS) both shortened to 13 min, which is only 7.1% lower than the IST and FST of specimen L-11. The dosage of LSS increased from 200 g to 300 g. Although the setting time still decreased, the decreasing range was significantly reduced. This is consistent with the effect of adding the same proportion of LSS (200 g to 300 g) on the development of compressive strength (only about 3%). Hence, the continuous acceleration of coagulation and significant improvement of mechanical properties cannot be achieved by continuously increasing the LSS content. The optimal improvement effect was achieved by adding 200 g LSS.

### 3.3. XRD Analysis

#### 3.3.1. XRD Analysis of PC Grouting Materials

The mineral crystal characteristics of PC and GBFS grouting materials at different proportions are shown in Figure 3. Figure 3a shows the XRD patterns of paste specimens L-1, L-3, and L-6 (1 d) in order to uncover the influence of pure cement, GBFS, and LSS on the mineral crystal characteristics of PC grouting materials. Clear characteristic peaks of portlandite mineral were found in specimen L-1, which implies the formation of Ca(OH)_2_. Ca(OH)_2_ is the main product of the hydration reaction of tricalcium silicate and the main component of PC. This shows that the hydration reaction of specimen L-1 was sufficient. The characteristic peak of quartz (SiO_2_) with high strength also appeared in specimen L-1 but it was not the product of the PC hydration reaction. Limestone and clay are required for the firing of PC. If they contain a large amount of quartz minerals (quartz does not easily react even during high-temperature calcination), these show up as characteristic peaks of quartz, which was the case in specimen L-1. Most importantly, the characteristic peaks of silicate minerals such as tobermorite and hillebrandite (representing the C-S-H gel) were found in sample L-1, but the intensity of the characteristic peak was low. The formation of C-S-H gel was clearly insufficient, which was the result of the poor crystallinity of the C-S-H gel and the short curing time (1 d) of specimen L-1. This was the key reason for the insufficient mechanical properties of PC grouting materials at an early age. Mineral characteristic peaks such as hydrogarnet and ettringite were also observed in sample L-1, and were the products of the tricalcium aluminate hydration reaction.

The addition of GBFS had an adverse effect on the mineral crystal structure of specimen L-3 (Figure 3a). Compared with specimen L-1, the intensity of the quartz characteristic peak in specimen L-3 did not significantly decrease, but the intensity of the portlandite characteristic peak decreased. This indicates that the formation of Ca(OH)_2_ was reduced to a certain extent and further shows that the degree of hydration reaction of PC decreased. The characteristic peak intensity of tobermorite and hillebrandite also decreased to varying degrees, indicating that the formation of hydrated calcium silicate minerals decreased. This also intuitively reflects the reduction of the degree of hydration reaction [45]. Moreover, the addition of GBFS did not result in new characteristic peaks for specimen L-3, indicating that no new minerals appeared. This confirms that GBFS can only be an inert filling in cement-based materials.

The addition of LSS did not significantly improve the characteristic peak intensities of tobermorite and hillebrandite in specimen L-6 (Figure 3a), compared with specimen L-3. The intensity of the portlandite characteristic peak did not decrease significantly. This shows that the added LSS did not react strongly with PC, nor did it adversely impact the hydration reaction. Most importantly, an albite characteristic peak appeared in specimen L-6, but it did not appear in specimens L-1 and L-3. This suggests that the addition of LSS was the main reason for the formation of albite. Due to the low degree of LSS participation in the hydration reaction, sodium and silicon introduced by LSS are in excess, and quickly undergo a self-condensation reaction to form albite minerals [46]. This reaction accelerates condensation and is also the reason for the slight increase in compressive strength. This further indicates poor synergy between LSS and PC.

#### 3.3.2. XRD Analysis of GBFS Alkali-Activated Grouting Materials

Figure 3b shows the XRD diffraction patterns of paste specimens L-8 and L-11 (1 d). The influence of sodium hydroxide content and LSS on the mineral crystal characteristics of GBFS alkali-activated grouting materials was analyzed. Compared with PC grouting materials, the mineral crystal characteristics of GBFS alkali-activated grouting materials were completely different. In addition to the characteristic peaks of silicate minerals such as tobermorite and hillebrandite observed in specimen L-8, the characteristic peaks of wollastonite mineral also appeared, which were not observed in PC grouting materials. This implies that silicate minerals also formed in GBFS alkali-activated grouting materials, and their types were more complex. Moreover, the unique characteristic peaks of gehlenite, anorthite, and scolecite, which did not appear in PC grouting materials, were also found in specimen L-8. Gehlenite, anorthite, and scolecite are polymerized from calcium, silicon, and aluminum. They are silicoaluminate minerals with similar properties as silicate minerals [47]. Therefore, the formation of aluminosilicate minerals was the key reason for the significantly faster setting of GBFS alkali-activated grouting materials compared to that of PC grouting materials and also their higher compressive strength.

With the addition of LSS, the mineral crystal characteristics of specimen L-11 (Figure 3b) were significantly improved compared with those of specimen L-8. The strength of the characteristic peak representing silicate minerals in specimen L-11 increased significantly, indicating that the crystallinity and silicate mineral formation were significantly improved. This proves that the crystallinity of silicate minerals improved, and that the formed amount increased significantly in the same curing time. Moreover, the strength of the characteristic peak representing aluminosilicate minerals in specimen L-11 also increased significantly, and new characteristic peaks for gehlenite appeared. All above-described phenomena show that the crystallinity of silicate and aluminosilicate minerals and the formation amount increased significantly after the addition of LSS. Therefore, LSS plays a synergistic role with GBFS alkali-activated grouting materials but it cannot do so with PC. As the polycondensation reaction of active substances must be carried out in the reaction solution, LSS creates a polymerization reaction environment that is rich in sodium and silicon [48]. This environment can accelerate the formation of minerals (especially aluminosilicate minerals), which was the reason for the accelerated condensation and significant increase in the compressive strength of specimen L-11.

### 3.4. SEM Analysis

#### 3.4.1. Microstructure of PC Grouting Materials

The microscopic reaction and microstructure characteristics of PC and GBFS grouting materials (both cured for 1 d) at different proportions are shown in Figure 4. The microstructure of specimen L-1 (100% PC) is shown in Figure 4a. Typical needle-like and ball-like hydrated calcium silicate gel products were observed, but the gel products were not tightly bonded and filled. Moreover, the microstructure showed many pores, and the diameter of pores varied from 1 to 20 µm. The reason was that at the initial stage of the PC hydration reaction, the hydration reaction rate was slow, and the formation of hydration products was insufficient. At the initial stage of microstructure growth and development, internal pores cannot be effectively filled, which was also the reason for the insufficient early compressive strength of specimen L-1.

Despite the addition of GBFS and LSS, the microstructure of specimen L-5 (Figure 4b) did not significantly improve. Spherical hydrated calcium silicate gel characteristics were still observed on the surface, but acicular hydrated calcium silicate gel did not appear, and there were no other shapes of gel products in specimen L-5. This shows that GBFS and LSS did not actively participate in the hydration reaction and formed new hydration products. Moreover, the increase in GBFS content led to a decrease in PC content, and further reduced the output of hydrated calcium silicate gel. This was also the reason for the disappearance of acicular calcium silicate gel [49]. More importantly, many micropores were still found in specimen L-5, and the diameter of pores was 1–15 µm, which was not significantly improved compared with specimen L-1. Therefore, the microstructure analysis indicates that the addition of GBFS and LSS cannot significantly improve the defects of PC grouting materials, which was the main reason why the compressive strength cannot be significantly increased.

#### 3.4.2. Microstructure of GBFS Alkali-Activated Grouting Materials

The microstructure of specimen L-8 (100% GBFS) is shown in Figure 4c. The surface not only showed the presence of spherical hydrated calcium silicate gel, but also sheet and block hydrated calcium aluminosilicate. Although specimen L-8 still contained pores and cracks, the number and diameter of pores were significantly lower compared with specimen L-1. Moreover, the microstructure surface of specimen L-8 was very flat, the bonding between microparticles was relatively tight, and the degree of polymerization was relatively high. This microstructure was completely different from the scattered and accumulated state of the surrounding particles in specimen L-1. This implies that the polymerization rate of GBFS alkali-activated materials was significantly faster than the hydration rate of PC, leading to a faster setting rate and better compressive strength of specimen L-8.

With the addition of LSS, the microstructure of specimen L-11 (Figure 4d) further improved significantly. Compared with specimen L-8, neither a spherical nor massive granular gel can be observed on the microsurface of specimen L-11. Although a few pores still appeared, the overall degree of polymerization was significantly improved. This shows that after the same curing time (1 d), LSS further accelerated the polymerization rate, improved the degree of polymerization, and further improved the microstructure [50,51]. Consequently, the condensation of specimen L-11 was further accelerated and its compressive strength was significantly improved. Therefore, LSS and GBFS can achieve an excellent synergistic reinforcement effect under highly alkaline conditions [52]. This can endow GBFS grouting materials with the characteristics of rapid setting (within 15 min) and adjustable setting time (the setting time can be adjusted within 15–30 min through different proportions), and also with the characteristics of early strength and high strength.

The laboratory grouting model experiment and the actual grouting process underground are shown in Figure 5.

## 4. Conclusions

In this study, PC and GBFS alkali-activated grouting materials were prepared with different mix proportions. XRF was carried out to determine the content and reactivity of various elements in PC and GBFS. The influence of different proportions on the development of compressive strength and condensation behavior was analyzed. The effects of sodium hydroxide and LSS on the formation of mineral crystals in both PC and GBFS alkali-activated specimens were further analyzed using XRD. The microstructure and morphological characteristics were analyzed by SEM.

The early compressive strength of cement grouting material was clearly insufficient. Moreover, both the strength development and setting were slow, making it difficult to achieve early reinforcement and the water plugging effect. Cement grouting material was especially unsuitable for use as grouting material in rescue engineering. Although the addition of LSS significantly accelerated the setting, it could not significantly improve the compressive strength. XRD analysis identified the formation of albite as the cause of the accelerated condensation.The early compressive strength of GBFS alkali-activated grouting material was significantly better than that of PC specimens (1 d strength increased by 120%), and its setting time was also significantly faster (IST shortened by 33.3%). With an increase in sodium hydroxide content, the compressive strength first increased and then decreased (maximum increase of 21.1%), while the setting time continued to shorten. XRD analysis identified the formation of aluminosilicate minerals as the main reason for the excellent mechanical properties and accelerated coagulation rate. SEM analysis showed that the crystal structure of aluminosilicate minerals was flat and smooth, making the microstructure denser and more complete.Based on GBFS alkali-activated grouting material, the addition of LSS can further improve the compressive strength and shorten the setting time. With an increase in LSS content, the compressive strength first increased significantly and then remained unchanged (maximum increase of 35.9%), while the setting time first decreased significantly and then remained unchanged (IST shortened by 58.1%). The optimal mix proportion was 40 g sodium hydroxide and 300 g LSS. XRD analysis showed that the crystallinity and formation of silicate and aluminosilicate were significantly improved after the addition of LSS. SEM analysis showed that the degree of polymerization after LSS addition and the microstructure of the specimen were significantly improved.This paper realized the preparation of early and high-strength slag alkali-activated grouting materials. However, extensive and in-depth research on hardness, fracture toughness, and impact strength is still needed to ensure their performance stability in all aspects.

## Figures and Tables

**Figure 1 polymers-14-03980-f001:**
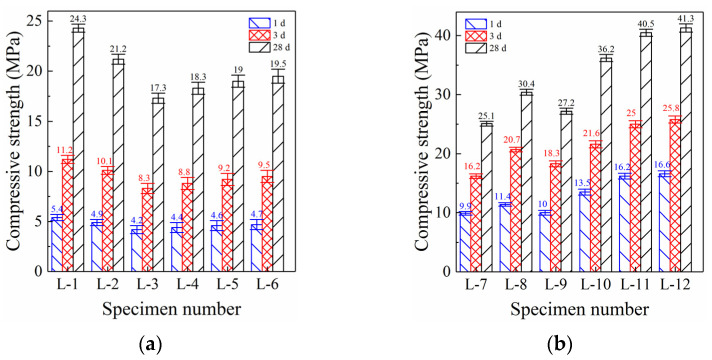
Development trend of compressive strength. (**a**) Portland cement grouting specimens. (**b**) GBFS alkali-activated grouting specimens.

**Figure 2 polymers-14-03980-f002:**
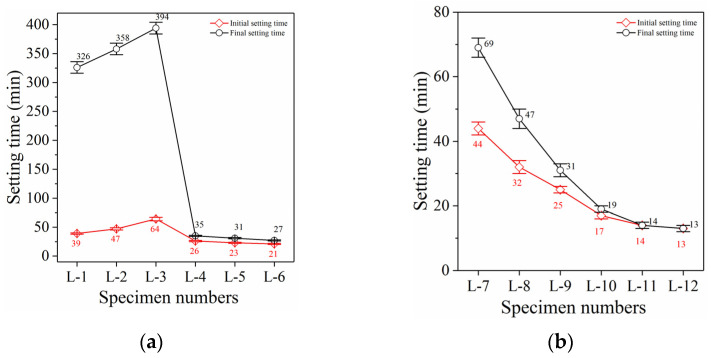
Analysis of condensation behavior. (**a**) Portland cement grouting specimens. (**b**) GBFS alkali-activated grouting specimens.

**Figure 3 polymers-14-03980-f003:**
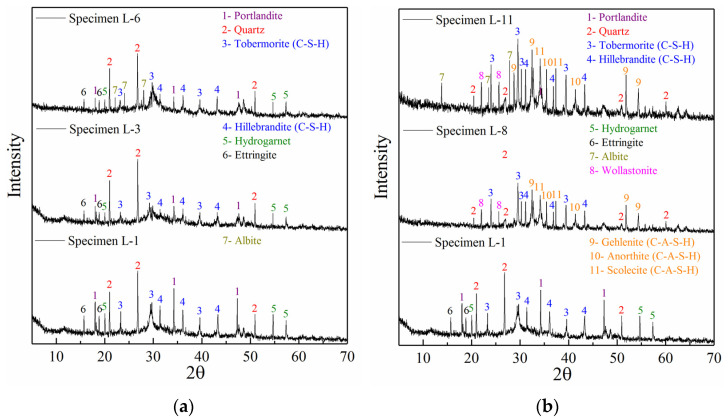
Crystal structure analysis of PC and GBFS alkali-activated grouting specimens. (**a**) PC grouting specimens. (**b**) GBFS alkali-activated grouting specimens.

**Figure 4 polymers-14-03980-f004:**
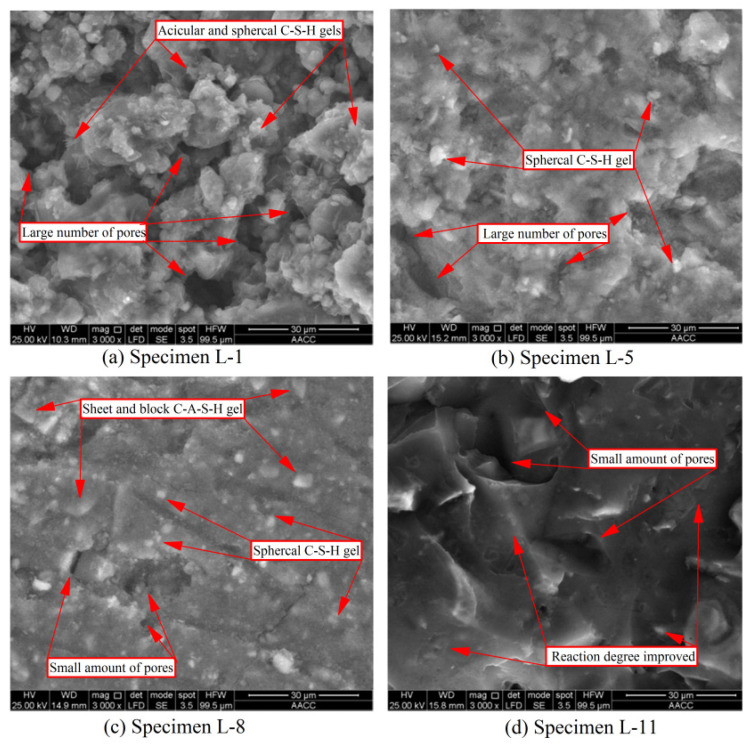
SEM analysis of grouting specimens.

**Figure 5 polymers-14-03980-f005:**
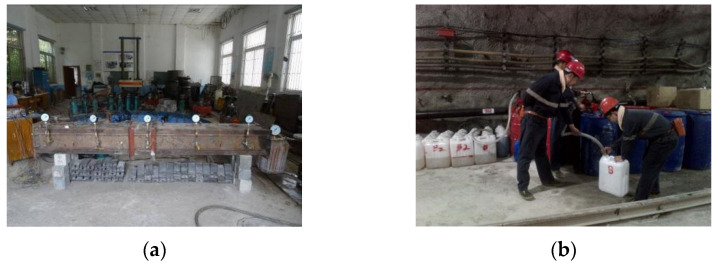
Laboratory grouting model experiment and actual grouting process underground. (**a**) Laboratory grouting model. (**b**) Actual underground grouting batching. (**c**) Carrying out underground grouting. (**d**) Grouting orifice.

**Table 1 polymers-14-03980-t001:** Chemical composition of Portland cement %.

Raw Material	SiO_2_	Al_2_O_3_	Fe_2_O_3_	CaO	MgO	Na_2_O	K_2_O	SO_3_	Others	Loss
PC	21.15	4.79	2.12	61.82	2.55	0.67	0.24	2.35	1.52	2.21
GBFS	32.42	20.85	0.69	33.79	6.36	1.32	0.77	-	2.31	0.73

**Table 2 polymers-14-03980-t002:** Mix proportion of specimens/g.

	PC	GBFS	LSS	Sodium Hydroxide	Water	Liquid–Solid Ratio
L-1	1000	0	0	0	600	0.6
L-2	800	200	0	0	600	0.6
L-3	600	400	0	0	600	0.6
L-4	600	400	100	0	535	0.6
L-5	600	400	200	0	470	0.6
L-6	600	400	300	0	405	0.6
L-7	0	1000	0	20	600	0.6
L-8	0	1000	0	40	600	0.6
L-9	0	1000	0	60	600	0.6
L-10	0	1000	100	40	535	0.6
L-11	0	1000	200	40	470	0.6
L-12	0	1000	300	40	405	0.6

Note: the content of free water in LSS is 65%.

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
