# Peer review of "Coagulation Mechanism and Compressive Strength Characteristics Analysis of High-Strength Alkali-Activated Slag Grouting Material"

_polymers, 2022, doi:10.3390/polym14193980_

Round 1

Reviewer 1 Report (New Reviewer)

Authors has made the correction based on comment but only one thing suggest to improve.

From the abstract, author mention 'X-ray diffraction analysis identified the formation of aluminosilicate minerals as the main reason for the excellent mechanical properties and accelerated coagulation rate.' - But in discussion, author not discuss details. Please add some Critical discussion on this.

Author Response

 Response to the Reviewers

The authors would like to thank all the reviewers for their great comments on the manuscript. Their comments have been taken into consideration seriously while revising the manuscript. The revisions are underlined in the revised manuscript for easier tracking. Their comments/concerns are addressed as follows. Note that the line numbers referred to by the reviewers may change due to the revision of the manuscript.

Reviewers' comments:

Reviewer #1:

  1. From the abstract, author mention 'X-ray diffraction analysis identified the formation of aluminosilicate minerals as the main reason for the excellent mechanical properties and accelerated coagulation rate.' - But in discussion, author not discuss details. Please add some Critical discussion on this.

Page 13, line 407-413 and 419-423, through XRD analysis, the author found that the characteristic peaks of gehlenite, anorthite, and scolecite appeared in specimen L-8, which these minerals are composed of calcium, silicon and aluminum and can be considered as aluminosilicate minerals. Moreover, XRD analysis shows that aluminosilicate minerals are not formed in cement grouting specimens. Furthermore, the experimental study on compressive strength and setting law shows that the compressive strength and setting time of alkali activated grouting specimens are significantly better than those of cement grouting specimens. Therefore, it can be concluded that the formation of aluminosilicate is due to the excellent compressive strength of alkali activated grouting materials and the shortened setting time.

Reviewer 2 Report (New Reviewer)

Manuscript ID: polymers-1898726

Title: Coagulation mechanism and compressive strength characteristics analysis of high strength alkali activated slag grouting material

Journal: Polymers

Comments to authors:

In this study, a type of alkali activated grouting material is prepared using slag as raw material combined with sodium hydroxide and liquid sodium silicate excitation. The compressive strength of specimen cured for 1d, 3d and 28 d is regularly measured and the condensation behavior is analyzed. Using X-ray diffraction and scanning electron microscopy, both formation behavior of mineral crystals and microstructure characteristics are further analyzed. This is well-written manuscript with limited novelty. However, to improve the quality of the manuscript, please address the following comments:

1)      The problem to be addressed in this study should also be highlighted in the Abstract.

2)      Please highlight the novelty in the Abstract.

3)      English proofreading is required for some grammatical mistakes and typos.

4)      The authors should also present some quantitative/numerical results in the Abstract.

5)      The novelty and significance of the present work should be highlighted in the last paragraph of the Introduction section.

6)      The authors are recommended to add latest relevant literature review on such works.

7)      What is the need for this work?

8)      Is this work helpful for practical applications?

9)      The literature review should be improved by adding latest references and discussion.

10)  Work methodologies need more discussion.

11)  Experimental proofs of the work should be given by adding images.

12)  Results section should be defended using technical reasons and relevant references

13)  SEM images need to be clearly discussed.

14)  More properties of produced grouting material such as hardness, fracture toughness, impact strength can be added.

15)  More technical discussion to the presented experimental results should be added.

16)  Conclusions should be refined and briefly presented. More numerical results should be added.

17)  The authors can add the future recommendations based on the present study.

Author Response

 Response to the Reviewers

The authors would like to thank all the reviewers for their great comments on the manuscript. Their comments have been taken into consideration seriously while revising the manuscript. The revisions are underlined in the revised manuscript for easier tracking. Their comments/concerns are addressed as follows. Note that the line numbers referred to by the reviewers may change due to the revision of the manuscript.

Reviewers' comments:

Reviewer #2:

In this study, a type of alkali activated grouting material is prepared using slag as raw material combined with sodium hydroxide and liquid sodium silicate excitation. The compressive strength of specimen cured for 1d, 3d and 28 d is regularly measured and the condensation behavior is analyzed. Using X-ray diffraction and scanning electron microscopy, both formation behavior of mineral crystals and microstructure characteristics are further analyzed. This is well-written manuscript with limited novelty. However, to improve the quality of the manuscript, please address the following comments:

1.The problem to be addressed in this study should also be highlighted in the Abstract.

It has been revised. Page 1, line 19-20.

  1. 2.Please highlight the novelty in the Abstract.

It has been revised. Page 1, line 18-19.

  1. English proofreading is required for some grammatical mistakes and typos.

It has been polished by professional translation agencies.

  1. 4.The authors should also present some quantitative/numerical results in the Abstract.

It has been revised. Page 1, line 27 and 29.

  1. 5.The novelty and significance of the present work should be highlighted in the last paragraph of the Introduction section.

It has been revised. Page 3, line 83 to 86.

  1. 6.The authors are recommended to add latest relevant literature review on such works.

It has been revised. Page 19, line 562 and 574.

  1. 7.What is the need for this work?

With the expansion of mining scope and the increase of depth, unstable and extremely unstable broken coal and rock mass are increasing day by day. In order to control the deformation of coal and rock mass, effective control measures must be taken. However, the traditional cement-based grouting materials have the defects of insufficient early strength and slow strength development, which can not meet the reinforcement of unstable and extremely unstable broken coal and rock mass. Therefore, it is necessary to develop early strength and high-strength grouting materials to meet the reinforcement needs of unstable and broken coal and rock masses.  

  1. Is this work helpful for practical applications?

With the expansion of mining scope and the increase of depth, unstable and extremely unstable broken coal and rock mass are increasing day by day. In order to control the deformation of coal and rock mass, effective control measures must be taken. However, the traditional cement-based grouting materials have the defects of insufficient early strength and slow strength development, which can not meet the reinforcement of unstable and extremely unstable broken coal and rock mass. However, alkali activated materials have unique characteristics of rapid solidification, high early strength and rapid strength development. Therefore, author introduces alkali activated materials into grouting materials to make up for the shortcomings of cement grouting materials in performance and enhance the reinforcement effect.

  1. 9.The literature review should be improved by adding latest references and discussion.

References [7] - [17], [21] [28], [30], [39] - [41], [43], [44], [51], [52], a total of 21 literatures are in 2020 to 2022, accounting for 40.4% of the total literatures, which latest relevant literature review is enough.  

  1. 10.Work methodologies need more discussion.

First of all, the compressive strength test and setting law test show that alkali activated grouting material is superior to cement grouting material in early compressive strength and setting time. Moreover, proper amount of sodium hydroxide and LSS can further optimize the performance of alkali activated grouting materials. However, it is difficult to explain the reason in mechanism why alkali activated grouting material has better performance only through macroscopic experimental data. Therefore, in Sections 3.1 and 3.2, only the experimental results are qualitatively discussed, but the mechanism is not analyzed due to lack of necessary microscopic experimental results to prove.

Then through XRD analysis, it is concluded that not only silicate minerals but also aluminosilicate minerals can be formed in alkali activated grouting materials, but aluminosilicate minerals are not formed in cement grouting materials. Moreover, proper amount of sodium hydroxide and liquid sodium silicate can also significantly promote the formation of silicate and aluminosilicate. Therefore, author concludes that the formation of aluminosilicate is the main reason why the compressive strength and setting time of alkali activated grouting materials are better than those of cement grouting materials, and discusses in detail the formation mechanism of aluminosilicate and its role (400-429). Finally, the conclusion of XRD analysis was further confirmed by SEM analysis.

Therefore, author did not carry out mechanism analysis on the experimental results of compressive strength and coagulation law, but used XRD and SEM analysis to prove the experimental conclusions, and then carried out full and thorough mechanism analysis along with attach necessary references. This makes the mechanism analysis between compressive strength and coagulation law less, which made the reviewers thought that the discussion was insufficient, but this is properly expounded in the microscopic analysis.

  1. 11.Experimental proofs of the work should be given by adding images.

It has been revised. Page 16, line 480-486.

  1. 12.Results section should be defended using technical reasons and relevant references

First of all, the compressive strength test and setting law test show that alkali activated grouting material is superior to cement grouting material in early compressive strength and setting time. Moreover, proper amount of sodium hydroxide and LSS can further optimize the performance of alkali activated grouting materials. However, it is difficult to explain the reason in mechanism why alkali activated grouting material has better performance only through macroscopic experimental data. Therefore, in Sections 3.1 and 3.2, only the experimental results are qualitatively discussed, but the mechanism is not analyzed due to lack of necessary microscopic experimental results to prove.

Then through XRD analysis, it is concluded that not only silicate minerals but also aluminosilicate minerals can be formed in alkali activated grouting materials, but aluminosilicate minerals are not formed in cement grouting materials. Moreover, proper amount of sodium hydroxide and liquid sodium silicate can also significantly promote the formation of silicate and aluminosilicate. Therefore, author concludes that the formation of aluminosilicate is the main reason why the compressive strength and setting time of alkali activated grouting materials are better than those of cement grouting materials, and discusses in detail the formation mechanism of aluminosilicate and its role (400-429). Finally, the conclusion of XRD analysis was further confirmed by SEM analysis.

Therefore, author did not carry out mechanism analysis on the experimental results of compressive strength and coagulation law, but used XRD and SEM analysis to prove the experimental conclusions, and then carried out full and thorough mechanism analysis along with attach necessary references. This makes the mechanism analysis between compressive strength and coagulation law less, which made the reviewers thought that the discussion was insufficient, but this is properly expounded in the microscopic analysis.

  1. 13.SEM images need to be clearly discussed.

First, lines 436 to 438 describe the microstructure characteristics of specimen L-1, and lines 439 to 422 analyze the causes of microstructure. The micro morphological characteristics of GBFS and LSS modified specimen L-5 are described in lines 444 to 446, and the reasons for the phenomenon are decomposed in lines 447 to 451.

Moreover, on lines 457 to 462 and lines 469 to 471, analyze the micro morphological characteristics of GBFS and LSS modified specimens L-8 and L-11, which on lines 463 to 465 and lines 471 to 478 further explained the mechanism of improvement by GBFS and LSS.

Therefore, the author has compared and analyzed the microstructure and morphology of cement grouting specimens L-1 and L-5 along with GBFS alkali activated grouting specimens L-8 and L-11, which further analyzed and expounded the improvement reasons and strengthening mechanisms.

  1. 14.More properties of produced grouting material such as hardness, fracture toughness, impact strength can be added.

This paper mainly studies the effects of sodium hydroxide and liquid sodium silicate on the mechanical properties and setting laws of alkali activated grouting materials, and mainly expounds the characteristics of early strength, high strength and rapid setting of alkali activated grouting materials. However, the hardness, fracture toughness, impact strength and other characteristics proposed by the reviewer are also very important, which will be discussed in detail in the following papers.

  1. 15.More technical discussion to the presented experimental results should be added.

The author integrates the experimental results with the discussion, and does not list a separate chapter to analyze the experimental results, which has fully demonstrated the experimental results in the original text.

For example, section 3.1.2 "compressive strength analysis", on lines 215 to 218, author first expounds the experimental data of specimen L-7 and compares it with specimen L-1. Then on lines 219 to 225, author summarizes the experimental results and draws corresponding conclusions. However, the mechanism of the phenomenon cannot be deeply analyzed and confirmed only by the macroscopic compressive strength experimental data.

Therefore, author only proves the conclusions in the compressive strength analysis through the formation of mineral crystals and the improvement of microstructure through the phenomena obtained in section 3.3.2 (XRD analysis) and section 3.4.2 (SEM analysis) and elaborates the improvement mechanism in depth on lines 405 to 412 and 461 to 461.

Similarly, the improvement effect of sodium hydroxide and LSS on the compressive strength of alkali activated grouting materials is in lines 226 to 247. The mechanism is explained by microscopic analysis at lines 413 to 428 and 468 to 478.

  1. 16.Conclusions should be refined and briefly presented. More numerical results should be added.

It has been revised. Page 17, line 494-507.

  1. The authors can add the future recommendations based on the present study.

It has been revised. Page 17, line 510-512.

Round 2

Reviewer 2 Report (New Reviewer)

This work can be accepted now.

This manuscript is a resubmission of an earlier submission. The following is a list of the peer review reports and author responses from that submission.

Round 1

Reviewer 1 Report

This study investigated the coagulation mechanism and compressive strength characteristics analysis of high strength alkali activated slag grouting material. The authors must address the following comments carefully. If not, I will consider reject the manuscript.

1.      I cannot find any content about ‘polymer’ in the manuscript. Is the manuscript in the scope of the Journal?

2.      The alkali activated materials are prepared using slag, sodium hydroxide, and liquid sodium silicate. The slag, sodium hydroxide, and liquid sodium silicate are all common raw materials used for preparing alkali activated materials. Their mechanisms on the strength, hydration, and durability have been widely analyzed in previous studies (please see the following references). The references are just a part. What is the innovate of this study? Please elaborate it in detail.

[1] Davoodabadi, M., Liebscher, M., Hampel, S. (2021). Multi-walled carbon nanotube dispersion methodologies in alkaline media and their influence on mechanical reinforcement of alkali-activated nanocomposites. Composites Part B-Engineering, 209, 108559.

[2] Xu, Z.K., Yue, J.C., Pang, G.H., et al. (2021). Influence of the Activator Concentration and Solid/Liquid Ratio on the Strength and Shrinkage Characteristics of Alkali-Activated Slag Geopolymer Pastes. Advances in Civil Engineering, 2021, 6631316.

[3] Tran, T.T., Kwon, H.M. (2018). Influence of Activator Na2O Concentration on Residual Strengths of Alkali-Activated Slag Mortar upon Exposure to Elevated Temperatures. Materials, 11(8), 1296.

3.      SEM and XRD are adopted to reveal the microscopic mechanisms. I don’t think they are enough, because the two methods both are not able to proceed the quantitative analysis. DSC is very important for the microscopic mechanisms.

4.      7-day strength is very important for the civil materials. Why is it missing in the study?

5.      The SEM and XRD are presented and analyzed under only one curing day. Why?  

6.      The technical parameters of the raw materials must be supplemented.

7.      The image quality of the SEM must be improved.

Author Response

 Response to the Reviewers

The authors would like to thank all the reviewers for their great comments on the manuscript. Their comments have been taken into consideration seriously while revising the manuscript. The revisions are underlined in the revised manuscript for easier tracking. Their comments/concerns are addressed as follows. Note that the line numbers referred to by the reviewers may change due to the revision of the manuscript.

Reviewers' comments:

Reviewer #1:

  1. I cannot find any content about ‘polymer’ in the manuscript. Is the manuscript in the scope of the Journal?

The concept of polymer was proposed by French David ovits in 1978. It is an inorganic polymer with three-dimensional three-dimensional network structure composed of AlO4 and SiO4 tetrahedral structural units. However, alkali activated materials mainly consider the polymerization reaction among calcium, silicon and aluminum. Therefore, polymer materials are considered as one kind of alkali activated materials.

  1. The alkali activated materials are prepared using slag, sodium hydroxide, and liquid sodium silicate. The slag, sodium hydroxide, and liquid sodium silicate are all common raw materials used for preparing alkali activated materials. Their mechanisms on the strength, hydration, and durability have been widely analyzed in previous studies (please see the following references). The references are just a part. What is the innovate of this study? Please elaborate it in detail.

2.1 The author has added these three articles to the references. Page 20, line 625-633.

2.2 In deep coal mining, grouting reinforcement and water blockage are the most effective means for reinforce extracting the rock mass of extremely broken coal. However, traditional cement grouting materials are not suitable for use in complex strata because of their insufficient early mechanical strength and slow setting time. However, slag alkali activated material has the characteristics of early strength, high strength and quick setting. Its application to grouting materials can solve the problems of insufficient early reinforcement strength and slow development of reinforcement strength, so that it can play a reinforcement role in deep, unstable and extremely unstable broken coal and rock mass, which cement grouting material cannot be realized.

  1. SEM and XRD are adopted to reveal the microscopic mechanisms. I don’t think they are enough, because the two methods both are not able to proceed the quantitative analysis. DSC is very important for the microscopic mechanisms.

XRD and SEM were used to analyze the microstructure, polymerization degree and the  mineral crystals characteristics in slag alkali activated grouting materials. Moreover, the influence of sodium hydroxide and LSS on the formation and transformation of mineral crystals and the improvement of the microstructure and polymerization degree were further analyzed. At the same time, combined with the compressive strength analysis, the most appropriate proportion of slag alkali activated grouting material is obtained and the mechanism is also analyzed.

The differential scanning calorimeter can analyze the polymerization reaction temperature of materials, and measure the phase transition temperature and thermal effect of materials. Of course, the crystallization of polymer materials can also be carried out. However, due to the defects of insufficient early strength and slow strength growth of cement grouting materials, this paper uses slag alkali activated materials in grouting materials to give full play to the characteristics of early strength and high strength. The mechanism of early strength and high strength of slag grouting materials is also analyzed by XRD and SEM.

  1.   7-day strength is very important for the civil materials. Why is it missing in the study?

Due to the deficiency of early strength compressive strength of traditional cement grouting materials and the slow development of early strength strength. This paper uses slag alkali activated materials in grouting materials to give full play to the characteristics of early strength and high strength, which improve the defects of poor early strength reinforcement effect of cement grouting materials. Therefore, the compressive strength of the specimens after curing for 1d and 3d was mainly tested, which represents the development law of early compressive strength. The 28 d mainly represents the later compressive strength. Therefore, the measurement of 7 d compressive strength was omitted.

  1. The SEM and XRD are presented and analyzed under only one curing day. Why? 

 In order to analyze the characteristics of early strength and high strength of slag alkali activated grouting materials, XRD and SEM were respectively used to analyze the mineral crystal characteristics and micro morphology characteristics of 1 d curing, and compared with cement grouting materials, so as to analyze the mechanism of early strength and high strength of slag alkali activated grouting materials.

  1. The technical parameters of the raw materials must be supplemented.

 The particle fineness and specific chemical composition of PC and GBFS have been discussed in detail in sections 2.1.1 and 2.1.2. Other technical parameters also comply with relevant national standards (GB175-2020) and (GB / T 18046-2017). Therefore, other technical parameters related to PC and GBFS which have little influence on compressive strength and setting law are not discussed in detail in this article.

  1. The image quality of the SEM must be improved.

Due to the limitations of school experimental conditions, this paper adopts Hitachi flexSEM 1000 scanning electron microscope (Hitachi, Tokyo, Japan). This is the highest resolution that this instrument can achieve under the condition of 3000 times magnification. It can completely distinguish the structural characteristics of microscopic particles and the shape characteristics of reaction products. Of course, more advanced detection equipment can get more beautiful pictures, but the current form can already show the microstructure characteristics.

Reviewer 2 Report

This is the review of the manuscript polymers-1872032, namely "Coagulation mechanism and compressive strength characteristics analysis of high strength alkali activated slag grouting material " by Mingjing Li, Guodong Huang, Yi Cui, Bo Wang, Shuwei Zhang, Qi Wang, Jiacheng Feng and Ming Ge 

General Comments:

In this paper, a high-performance grouting material for underground mines is prepared using slag as the main raw material and both sodium hydroxide and liquid sodium silicate as activators. This material is compared with traditional cement grouting material. The compressive strength development and condensation behavior of slag grouting materials are analyzed mainly focusing on the formation and transformation characteristics of mineral crystals in grouting materials. The application of slag alkali activated materials for the surrounding rock grouting of deep coal mines is an innovation that fully highlights the performance advantages of slag alkali activated materials.

The scientific research looks very good. I accept manuscript in present form.

Author Response

 Response to the Reviewers

The authors would like to thank all the reviewers for their great comments on the manuscript. Their comments have been taken into consideration seriously while revising the manuscript. The revisions are underlined in the revised manuscript for easier tracking. Their comments/concerns are addressed as follows. Note that the line numbers referred to by the reviewers may change due to the revision of the manuscript.

Reviewers' comments:

Reviewer #2:

In this paper, a high-performance grouting material for underground mines is prepared using slag as the main raw material and both sodium hydroxide and liquid sodium silicate as activators. This material is compared with traditional cement grouting material. The compressive strength development and condensation behavior of slag grouting materials are analyzed mainly focusing on the formation and transformation characteristics of mineral crystals in grouting materials. The application of slag alkali activated materials for the surrounding rock grouting of deep coal mines is an innovation that fully highlights the performance advantages of slag alkali activated materials.

Thank you.

Round 2

Reviewer 1 Report

The revised manuscript is very perfunctory. My comments are not addressed well.